# FINE-TUNED LANGUAGE MODELS GENERATE STABLE INORGANIC MATERIALS AS TEXT

**Nate Gruver**[1]  **Anuroop Sriram**[2]  **Andrea Madotto**[2]
**Andrew Gordon Wilson**[1]  **C. Lawrence Zitnick**[2]  **Zachary Ulissi**[2]
[1]NYU  [2]Meta FAIR

## ABSTRACT

We propose fine-tuning large language models for generation of stable materials. While unorthodox, fine-tuning large language models on text-encoded atomistic data is simple to implement yet reliable, with around 90% of sampled structures obeying physical constraints on atom positions and charges. Using energy above hull calculations from both learned ML potentials and gold-standard DFT calculations, we show that our strongest model (fine-tuned LLaMA-2 70B) can generate materials predicted to be metastable at about twice the rate (49% vs 28%) of CD-VAE, a competing diffusion model. Because of text prompting's inherent flexibility, our models can simultaneously be used for unconditional generation of stable material, infilling of partial structures and text-conditional generation. Finally, we show that language models' ability to capture key symmetries of crystal structures improves with model scale, suggesting that the biases of pretrained LLMs are surprisingly well-suited for atomistic data.

## 1 INTRODUCTION

Large language models (LLMs) are trained to compress large text datasets, but can also act as strong foundations for non-text data (Delétang et al., 2023). As compressors, LLMs extract common patterns and find simple programs that can produce them (Goldblum et al., 2023; Sutskever, 2023), regardless of the data's origin. From text pretraining alone, LLMs can compress or extrapolate data as diverse as images (Delétang et al., 2023), tabular data (Goldblum et al., 2023), time series (Gruver et al., 2023a), or robotic trajectories (Mirchandani et al., 2023). Alongside generality, LLM pre-training also gives rise to sample efficiency, as in-context learning and fine-tuning require far fewer training examples to identify salient patterns than training a model from scratch (Brown et al., 2020).

The generality and sample efficiency of LLMs make them particular promising for scientific problems, where data are often limited, collected from diverse sources, or challenging for non-experts to interpret. In materials science, for example, the number of known stable materials is relatively small, and the data describing each material are diverse, including composition, structure, and complex properties. LLMs can learn generalizable rules from a small number of examples (Zhu et al., 2023), combine modalities into a single model (Moon et al., 2023), and provide users with a text-based interface. A text interface, in particular, has the potential to improve access to scientific discovery (White, 2023); LLMs can use text to describe new observations, or, in design applications (e.g. materials design, drug discovery), LLMs can ingest text that specifies desired properties or constraints (Bran et al., 2023).

In this work, we show that fine-tuned LLMs can generate the three-dimensional structure of stable crystals as text (Figure 1). Our method is simple: first, encode crystals as new-line separated strings and combine with text instructions, then perform parameter efficient fine tuning (PEFT) on a base LLM (LLaMA-2) with a multitask curriculum and translation augmentations (Section 4). We evaluate our method with Materials Project data (Jain et al., 2013), comparing against an invariant diffusion model and a sequence model trained from scratch. Using both learned ML potentials and gold-standard DFT calculations, we show that our method can generate materials predicted to be stable at higher rates than baseline methods. To understand the success of our fine-tuning approach, we probe the learned symmetry properties of our model, proposing a new metric for language models trained on atomistic data and examining the effect of model scale on learned invariance. Going beyond

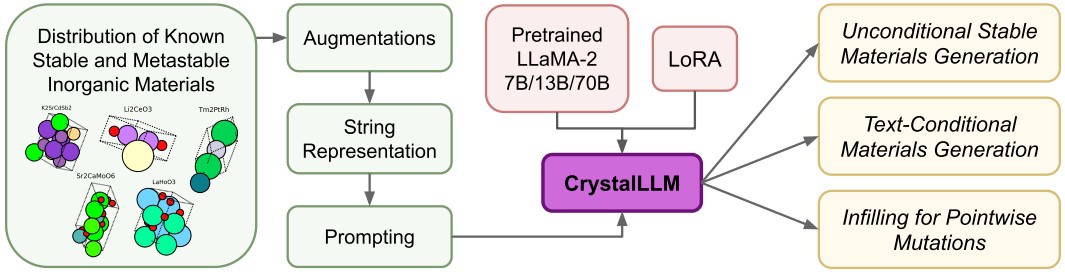

Figure 1: Overview of our approach to materials generation with large language models. Using string formatted crystals and task-specific prompting, we enable unconditional stable materials generation, text-condition materials generation, and structural infilling. Base LLaMA-2 models are fine-tuned on a database of known inorganic materials (Liu et al., 2020) using low-rank adapters.

unconditional generation, we also show that our LLMs have other useful abilities within materials design, such as text-conditional generation and infilling, which can be used to optimize the properties of existing materials.[1]

## 2  RELATED WORK

There are two central challenges in applying generative models to crystals and related atomistic data. The first challenge is that atoms are intrinsically both discrete and continuous objects, as each atom has both an element identity and a position in three dimensional space. Approaches to generative modeling often differ between for discrete and continuous data, and modeling both simultaneously can be significantly more complex than modeling either individually. The second key challenge is the prevalence of symmetries in atomistic data. The unit cell, a repeated pattern tiled infinitely in every direction, is the common representation for crystals because it easily captures translation invariance, the fact that atoms can be shifted and wrapped around the unit cell while still representing the same underlying structure. Symmetries can pose challenges to deep learning models because they entail constraints on the functions that neural networks can learn.

**Diffusion models**  Xie et al. (2021) introduced crystal diffusion variational autoencoder (CDVAE) to directly deal with both of these challenges. CDVAE uses several individual generative models for discrete and continuous components that share a continuous (VAE) latent space. The chemical composition is reconstructed from this latent space using a language modeling head, while atom positions are generated with a denoising diffusion model (Ho et al., 2020). Since CDVAE, several works have extended diffusion processes to capture all parameters of the crystal, not just the atomic coordinates. Both Jiao et al. (2023) and Zeni et al. (2023) accomplish this by creating diffusions for the lattice parameters and atom identities, while Yang et al. (2023) design a new continuous representation that unifies atom identities and positions in a single high-dimensional tensor. In most cases, these diffusion models were designed with a careful eye towards symmetries and are built on top of graph neural networks with strict invariance/equivariance properties (Xie et al., 2021; Jiao et al., 2023; Zeni et al., 2023). The approach of Yang et al. (2023) is more similar to ours, as they apply a general-purpose architecture (3D U-net) and modeling approach (Gaussian diffusion) to a new representation, without guaranteeing symmetries. Discrete atom identities and variable length (number of atoms), however, require special considerations in diffusion models, unlike standard language models, which were originally designed for modeling discrete sequences.

**Language models**  Flam-Shepherd & Aspuru-Guzik (2023) demonstrate an alternative to continuous denoising models and architectural invariances. Instead of treating discrete and continuous modalities separately, as in CDVAE, Flam-Shepherd & Aspuru-Guzik (2023) uses sequences of discrete tokens to represent everything, including the digits of atomic coordinates. With all data encoded as tokens, standard language modeling methods designed for text can be applied with little to no modification. The simplicity of this method also makes it simple to adapt to many different kinds of molecular

---

[1]https://github.com/facebookresearch/crystal-llm

structures, including small molecules, protein binding pockets, and, of course, crystals. In lieu of architectural symmetries, augmentations of the training data are used to encourage learning known invariances. Flam-Shepherd & Aspuru-Guzik (2023) demonstrates that language models trained from scratch on many common molecular datasets actually outperform popular domain-specific models, including CDVAE, in their ability to capture valid element compositions and high-level statistics of the training data. Similarly, Antunes et al. (2023) also use language models to generate crystal structures as discrete sequences by training from scratch on millions of CIF strings.

**Our work** In this work, we show that pretrained LLMs are also useful for understanding and generating 3-dimensional atomic structures. By using a pre-trained LLM, we can achieve high rates of validity without crystal-specific tokenization (Flam-Shepherd & Aspuru-Guzik, 2023) or millions of auxiliary structures (Antunes et al., 2023). Unlike many methods designed specifically for crystal structures and symmetries, our method can also be easily extended to multiple crystal generation tasks and, in the future, to other atomistic modalities without any changes to the underlying model or training procedure. Building on the basic observations made by Flam-Shepherd & Aspuru-Guzik (2023), we show that larger models, which are often more effective compressors of data, demonstrate improved ability to learn symmetries from the training data and augmentation.

## 3 BACKGROUND

**Language Modeling** LLMs perform next-token prediction over sequences. The model is a categorical distribution, $p(w_{t+1}|w_{0:t})$, where $w_{0:t}$ is the prompt, a sequence of input tokens, and $w_{t+1}$ is the predicted next token. To generate sequences from the model, the conditional distribution is sampled sequentially, but samples are rarely drawn from the original, unmodified categorical distributions. Instead the sampling procedure is typically modulated with *temperature* ($\tau$) and *nucleus size* ($p$) hyperparameters. Temperature serves to flatten the conditional distributions to uniform (high temperature) or collapse them around their maximal probabilities (low temperature). Nucleus size limits which tokens can be sampled based on the cumulative distribution function, clipping out values that contribute very little mass. A nucleus of $p$ $(0 < p \leq 1)$ corresponds to keeping tokens to cumulatively contribute $p\%$ of the total probability, and discarding the rest.

**Tokenization** To train language models on text datasets, strings are converted into sequences of tokens. Most modern LLMs rely on *byte pair encoding* (BPE) (Gage, 1994), a compression method that assigns tokens to common substrings, making overall sequence lengths shorter. One downside of BPE tokenization is the default tokenization of numbers. BPE typically breaks numbers into irregular substrings instead of individual digits. While breaking numbers into multi-digit tokens creates shorter sequences, it also complicates learning basic arithmetic operations, which typically operate at the level of individual digits. Luckily, Touvron et al. (2023b) introduce tokenizers for LLaMA-2 models that break numbers into a sequence of digits, which has been shown to dramatically improve performance on arithmetic tasks (Liu & Low, 2023). We use LLaMA models in our work because they have a natural representation of 3D coordinates and can therefore learn simple functions over those coordinates that obey domain-specific symmetries (Section 5).

**Crystal structures and energy prediction** Periodic materials are defined by a unit cell repeated infinitely along all three dimensions (Figure 2). The unit cell comprises a lattice (parallelepiped) with side lengths ($l_1, l_2, l_3$) and angles ($\theta_1, \theta_2, \theta_3$). Within the lattice, there are $N$ atoms, each specified by an element identity, $e_i$, and set of 3d coordinates ($x_i, y_i, z_i$) which can be absolute or fractional (specified as a percentage of the unit cell side lengths). Therefore a bulk material can be fully described by the tuple

$$C = (l_1, l_2, l_3, \theta_1, \theta_2, \theta_3, e_1, x_1, y_1, z_1, ..., e_N, x_N, y_N, z_N). \tag{1}$$

For a given set of environmental conditions, every crystal has a corresponding energy that describes how likely it will occur in a particular configuration. Configuration with unfavorable electrostatic interactions from unlike atomic positions, such as highly overlapping atoms, are typically high energy. The gold standard for energy prediction is density functional theory (DFT), which provides tractable approximations to governing quantum mechanical equations that describe the energy and time evolution of a system. DFT, however, can be prohibitively expensive, often scaling $O(n^3)$ with the system size, which has motivated development of deep learning potentials to approximate DFT solutions (Lan et al., 2022).

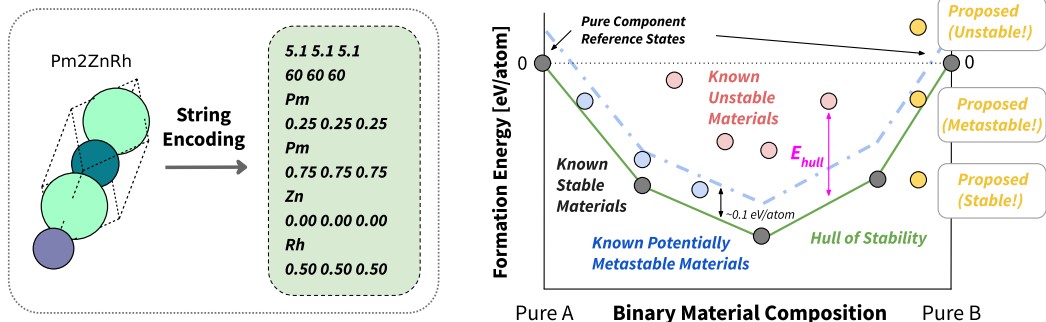

Figure 2: (**left**) We convert the crystal lattice, atom identities, and atom positions into strings. The model is trained to generate a structures conditioned on the text prompt, which might contain additional information about the composition, properties, or a starting structure to modify. (**right**) Energy above hull ($E_{\text{hull}}$) quantifies the stability of a material. A crystal with $E_{\text{hull}} < 0.1$ will be energetically favorable both in its structure and composition.

**Stability of hypothetical materials** ($E_{hull}$)   The composition of a crystal also impacts its energy, as different elements have different geometries and charge properties. Certain stoichiometries, or ratios of elements, are naturally favored, and a composition of elements $A$ and $B$ with constituent parts $A_x B_y$ can dissociate into the composition $A_c B_d$ if it is energetically favorable. Because of the effect of composition, the energy of a crystal is typically a two dimensional concept captured by the energy hull, which is the minimum observed configuration energy for a given composition. For a crystal to be low-energy and stable, and therefore give rise to a practically useful material, it must have a small *energy above hull ($E_{hull}$)*, the distance from the energy hull for the crystals elemental composition (Figure 2). Crystals with $E_{\text{hull}} < 0$ are considered stable and by definition have lower energy than the known minimum (which has $E_{\text{hull}} = 0$). Crystals with $E_{\text{hull}} < 0.1$ eV/atom are often *metastable* and likely to be practical useful (Sun et al., 2016).

## 4   METHOD

Our approach to generating stable materials is pleasingly simple. We take a pre-trained LLM, which has useful biases towards generalizable patterns, and fine-tune it on crystal string representations. Because language models can also ingest text, we can condition the model's generations on text descriptions. The flexibility of language models also allows us to solve other tasks, such as infilling, through small modifications to the input formatting. Though we focus solely on crystal structures in this work, our method itself is general purpose and could be easily extended to proteins, nucleic acids, or small molecules. We include a more detailed discussion of how general text-pretraining impacts our method in Appendix A.5.

**String formatting and tokenization**   We convert the crystal tuple $C$ (Equation 1) using fixed precision numbers. An example of crystal string formatting is shown in Figure 2. We represent lattice lengths with one decimal place (2-3 digits) and lattice angles as integers (1-3 digits). Fractional coordinates are always represented with two digits. 3D coordinates are combined with spaces and all other crystal components are combined with newlines. We deliberately chose LLaMA-2 models because they are both state-of-the-art in overall performance among open-source models and because they tokenize numbers as individual digits by default. Notably, it is therefore impossible to create one token per full number, as Flam-Shepherd & Aspuru-Guzik (2023) do in their best performing model (further discussion in Appendix A.1). Instead, we rely on the extensive pretraining of LLaMA-2 models to instill useful biases over numerical operations (Liu & Low, 2023).

**Prompt design**   To train a model that can be used for many tasks, including unconditional generation, text-conditional generation, and infilling, we use task-specific prompts. The input to the model is a prompt followed by the string-formatted crystal (Figure 2). In the most basic case, the prompt indicates that the model should generate bulk materials represented as a lattice and atoms. The prompt can also be expanded to include a desired composition or material properties, or to include a starting

structure, in the case of infilling. For infilling, the prompt includes the string-formatted crystal with every instance of a randomly chosen element replaced with [MASK], and the model is trained to generate the identity of the masked element at the end of the sequence. During training all three tasks are included through random sampling, with two thirds generation and one third infilling (details in Appendix A.2). As in instruction tuning, the prompt is given as input to the model but does not contribute to the generative loss function. The model is only penalized for its predictions on the crystal string or masked element.

| Generation Prompt | Infill Prompt |
|---|---|
| \<s\>Below is a description of a bulk material. [The chemical formula is Pm2ZnRh]. Generate a description of the lengths and angles of the lattice vectors and then the element type and coordinates for each atom within the lattice:

[ Crystal string ]\</s\> | \<s\>Below is a partial description of a bulk material where one element has been replaced with the string "[MASK]":

[ Crystal string with [MASK]s ]

Generate an element that could replace [MASK] in the bulk material:

[ Masked element ]\</s\> |

Blue text is optional and included to enable conditional generation. Purple text stands in for string encodings of atoms.

**Augmentations**   Crystals structures are symmetric under translational. All atomic coordinates can be shifted modulo the lattice boundaries without changing the resulting material structure. Similarly, the ordering of atoms within the lattice is irrelevant to the underlying material (permutation invariance). Prior work on diffusion generative models guarantee these symmetries as invariance or equivariance constraints on the model architecture (Xie et al., 2021; Jiao et al., 2023). To encourage translation invariance in our language models, we apply random uniform translations to the fractional coordinates. We chose not to augment the ordering of atoms because these variables often contained valuable information, for example grouping set of elements together for placement in the lattice (discussion in Appendix A.1).

## 5    EXPERIMENTS

We explore several uses of language models in crystal generative modeling. First, in order to compare with prior work, we show that fine-tuned LLMs can be used for unconditional generation of novel materials and that the resulting materials correspond to stable relaxed structures under the predictions of an ML potential and DFT. We then show that LLMs can also be used for text-conditional generation and to propose small changes to existing materials.

**Datasets and models**   For consistency with prior work (Xie et al., 2021; Flam-Shepherd et al., 2023) we used MP-20 (Jain et al., 2013), a dataset of 45231 materials, when training for unconditional generation. All structures in MP-20 are stable, and therefore an effective generative model trained on MP-20 should tend to propose new crystals that are at least metastable. For text-conditioned generation, we train with all forms of prompting (Section 4) on a collection of  120,000 crystals from Materials Project (Appendix A.3). The collection includes basic property information, such as the space group number, band gap, $E_{\text{hull}}$ and the chemical formula. All of our experiments were conducted with LLaMA-2 models (7B 13B, and 70B) (Touvron et al., 2023a;b) through the Transformers library (Wolf et al., 2020) and PyTorch (Paszke et al., 2019). In order to train on small number of GPUs we use 4-bit quantization (Dettmers et al., 2022) and Low-Rank Adapters (LoRA) (Hu et al., 2021). We provide the full hyperparameters and training details in Appendix A.4.

**Evaluation**   For basic evaluation of the LLM samples, we use the validity and diversity metrics introduced by Xie et al. (2021). Structural validity is determined by non-overlapping atomic radii (overlapping taken to be both atoms within half a radius of each other), while compositional validity captures the net charge of the structure (only structures with net neutral total charge are valid). Diversity is computed as pairwise distance between samples under featurizations of the structure and composition from Matminer (Ward et al., 2018; Xie et al., 2021).

Table 1: Following prior work (Xie et al., 2021), we evaluate fine-tuned LLaMA-2 models using validity, which captures physical constraints, as well as coverage and property metrics, which capture alignment between the ground truth and sampling distribution. We add stability checks, which count the percentage of samples estimated to be stable by M3GNet (Chen & Ong, 2022) and DFT (Hafner, 2008) (details in Appendix B.2). LLaMA models generate a high percentage of both valid and stable materials.

| Method | Validity Check | | Coverage | | Property Distribution | | Metastable | Stable |
|---|---|---|---|---|---|---|---|---|
| | Structural↑ | Composition↑ | Recall↑ | Precision↑ | wdist ($\rho$)↓ | wdist ($N_{el}$)↓ | M3GNet ↑ | DFT[†] ↑ |
| CDVAE | **1.00** | 0.867 | 0.991 | 0.995 | 0.688 | 1.43 | 28.8% | 5.4% |
| LM-CH | 0.848 | 0.835 | 0.9925 | 0.9789 | 0.864 | 0.13 | n/a | n/a |
| LM-AC | 0.958 | 0.889 | 0.996 | 0.9855 | 0.696 | 0.09 | n/a | n/a |
| **LLaMA-2** | | | | | | | | |
| 7B ($\tau = 1.0$) | 0.918 | 0.879 | 0.969 | 0.960 | 3.85 | 0.96 | 35.1% | 6.7% |
| 7B ($\tau = 0.7$) | 0.964 | 0.933 | 0.911 | 0.949 | 3.61 | 1.06 | 35.0% | 6.2% |
| 13B ($\tau = 1.0$) | 0.933 | 0.900 | 0.946 | 0.988 | 2.20 | 0.05 | 33.4% | 8.7% |
| 13B ($\tau = 0.7$) | 0.955 | 0.924 | 0.889 | 0.979 | 2.13 | 0.10 | **38.0%** | 14.4% |
| 70B ($\tau = 1.0$) | 0.965 | 0.863 | 0.968 | 0.983 | 1.72 | 0.55 | **35.4%** | 10.0% |
| 70B ($\tau = 0.7$) | **0.996** | **0.954** | 0.858 | 0.989 | 0.81 | 0.44 | **49.8%** | 10.6% |

[†] Fraction of structures that are first predicted by M3GNet to have $E_{\text{hull}}^{\text{M3GNet}} < 0.1$ eV/atom, and then verified with DFT to have $E_{\text{hull}}^{\text{DFT}} < 0.0$ eV/atom.

While useful for sanity checking models, simple validity metrics only reflect a subset of our real-world priorities in generating novel materials. Arguably the most important property that we hope to assess in samples is their predicted stability, which we can approximate by predicting the energy of relaxed structures. Using known materials and energy calculations from Materials Project we construct the ground truth energy convex hull and then calculate the approximate energy above hull, $\hat{E}_{\text{hull}}$. We chose two methods to estimate material stability:

- **ML potential**: M3GNet (Chen & Ong, 2022) provides energy, force, and stress approximations for crystal unit cells. For each sample we first run a relaxation using force and stress approximations then use the energy of the final structure.

- **DFT**: We run a relaxation using the Density Functional Theory code VASP (Hafner, 2008) with INCAR settings chosen by Pymatgen (Ong et al., 2013). DFT is the more accurate, but also much more computationally intense, of the two options.

In both cases, results are compatible with Materials Project values (Jain et al., 2013) (Appendix B.1). Because DFT is prohibitively expensive for many use cases (often hours per calculation), we only use it to double-check results obtained with ML potentials, and we only run VASP calculations on materials that have already been predicted as metastable by M3GNet (<0.1 eV/atom $\hat{E}_{\text{hull}}$). The use of a M3GNet surrogate model is not perfect as many structures in Figure 4 (right) have energies above the expected 0.1 eV/atom threshold, but the structures are largely close to the hull compared to the broader distribution of materials generated.

**Unconditional generation**   We sample 10,000 structures from each fine-tuned LLaMA model, parsing a CIF from the generated string. We reject the sample and draw another if a CIF cannot be parsed from the sampled string, which guarantees all samples can be interpreted as crystals but does not guarantee validity of the resulting crystal. We show the validity and predicted stability (Xie et al., 2021) of the resulting structures in Table 1, which shows that LLMs can achieve near-perfect rates of structural and compositional validity. Hyper-parameters like temperature and nucleus size can be used to trade-off validity and stability of samples with their coverage (Appendix B.3). LLaMA-2 70B strikes an effective balance, generating high rates of stable materials with good coverage and diversity (Figure 4). By default, generation is completely unconstrained and therefore the model can hallucinates imaginary elements, for example "Ln," a common abbreviation for Lanthanide (Figure 3), but the problem can be easily avoided by constraining the tokens for element identities (Wolf et al., 2020).

```
6.7 6.7 6.7
138.0 138.0 59.0
Ln
0.88 0.05 0.96
Ln
0.63 0.30 0.46
ln
0.38 0.55 0.96
ln
0.13 0.80 0.46
```

Figure 3: A sample with "hallucinated" element identities (Ln).

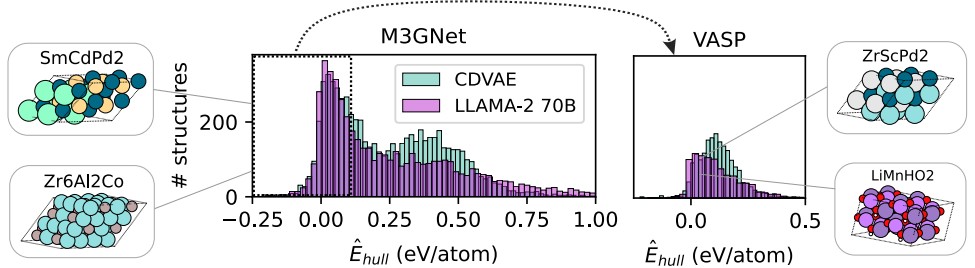

Figure 4: Stability of LLaMA samples compared to CDVAE (Xie et al., 2021). Fine-tuned LLaMA-2 70B generates a higher rate of metastable ($\hat{E}_{\text{hull}} < 0.1$) and stable materials than CDVAE, using estimates of $\hat{E}_{\text{hull}}$ from both M3GNet (Chen & Ong, 2022) and VASP (Hafner, 2008). Because of computational cost, we only run VASP on structures predicted to be stable by M3GNet. Stable materials generated by LLaMA are also more diverse (as quantified by Matminer featurization (Ward et al., 2018)) than stable samples from CDVAE. We include sampled stable structures, shown as (2,2,2) supercells, which display a high-degree of regularity and understanding of three-dimensional space.

**Symmetry learning**   As crystal structures have translational symmetry, ideally our model's likelihood should be invariant to translations. We propose *Increase in Perplexity under Transformation (IPT)* as metric for assessing the invariance of language models to continuous group transformations. For a transformation group $G$ with group elements $g$ and group action $t$, we define IPT for an input $s$,

$$\text{IPT}(s) = \mathbb{E}_{g \in G}[\text{PPL}(t_g(s)) - \text{PPL}(t_{g^*}(s))]$$

where

$$g^* = \arg\min \text{PPL}(t_{g^*}(s))$$

and PPL is the perplexity of the sequence, the exponent of the length-normalized cross entropy loss, $\text{PPL}(s) = 2^{\text{CE}(s)/n}$. In our case $G$ is the group of translation, where each $g$ is a distance to translate by, and $t_g$ is the mapping that decode the string, translates the coordinates (wrapping them around the boundary), and re-encodes the string. IPT captures the degree to which transformations change a language model's compression ability. Good understanding of group transformations and invariance in the data should lead to minimal change in the perplexity of a transformed sequence. We can approximate IPT by sampling many values of $g$ (e.g. 20), picking $g^*$ as the minimum among those values, and computing a sample mean. Figure 5 shows the mean IPT of 500 random crystals from the test set, for each of the three LLaMA model sizes. We include additional details about our IPT calculation in Appendix B.5.

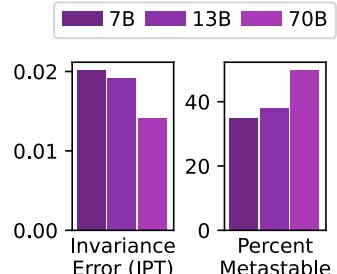

Figure 5: Translation invariance on test data and ability to generate stable materials increase in proportion. Larger models learn invariances from augmentations more effectively during training, likely as a result of their preference for abstract and compressible patterns.

**Diversity, novelty, and sampling speed**   When using generative models to discover new stable materials, there are several properties beyond the rate of stability that are practically significant. Novel and diverse samples encourage sufficient exploration of unknown material space, and sampling speed dictates how expensive it is to search within that space. We compare these properties for LLaMA-2 models and CDVAE in Figure 6. To calculate diversity and novelty, we use the same featurizations as in Table 1, calculating pairwise distances for diversity and distance to the closest neighbor in the training set for novelty (details in Appendix B.6). All metrics are computed over crystals judged metastable by M3GNet, so that all novelty and diversity are relevant and not an artifact of invalid generations. LLaMA-2 samples match or exceed the diversity of CDVAE samples and also obtain high rates of novelty when we consider both composition and structure. Interestingly, larger LLaMA models display less novel structures but more novel compositions. It's worth noting, however, that both CDVAE and LLaMA-2 7B far exceed the structural novelty of a held out test set, while 13B and 70B are just slightly lower. To judge sampling speed, we calculate the time required for 10,000

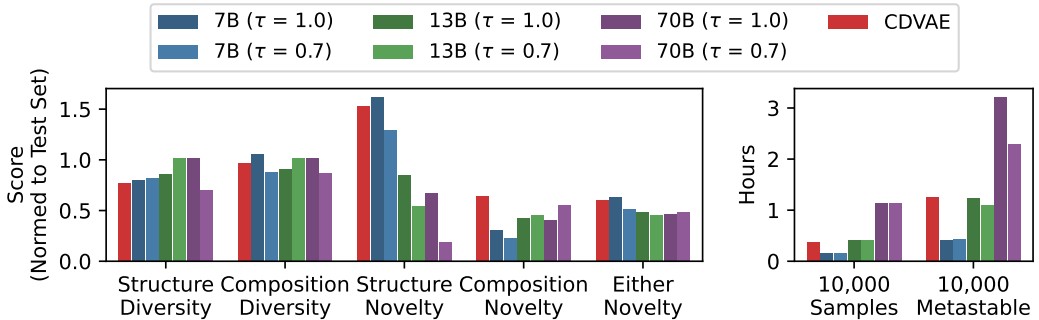

Figure 6: We compare LLaMA-2 models with CDVAE in their ability to generate novel and diverse samples as well as their overall speed. (**left**) We calculate diversity and novelty using a featurization of structure and composition (as in Table 1). Diversity is calculated as pairwise distance in feature space, while novelty quantifies the percentage of inputs that are far from the training set (Appendix B.6). All metrics are calculated only for samples that were already judged to be metastable. LLaMA-2 models often generate more diverse samples than CDVAE, and achieve similar overall rates of novelty. Interestingly, structural novelty is lower in larger models, while compositional novelty is higher. (**right**) We compare the time required to generate 10,000 samples from each model. We run LLaMA-2 models with the largest feasible batch size on one A100 GPU (Appendix B.7). While the largest LLaMA model is computationally expensive, smaller language models are very fast, especially when we consider both sampling speed and rate of stability.

samples, using the largest possible batch size on one A100 GPU (Appendix B.7). In Figure 6, we compare the sampling speed with CDVAE and find that smaller models are often significantly faster when generating metastable samples.

**Text-conditioned generation**    Extending our method to text-conditional generation is as simple as including additional information in the prompt, with a small amount of additional text (Figure 4). We explore conditioning on spacegroup number, composition, and $E_{hull}$, as these properties are easy to verify (at least approximately) in silico. We assess the model's ability to perform conditional generation by comparing the intended condition with labels obtained from an in-silico oracle for the constraint. For the chemical formula, we simply parse the composition from the generated CIF. For space group determination, we use pymatgen's SpacegroupAnalyzer with a precision of 0.2 angstroms (Ong et al., 2013). For stability, we use M3GNet to estimate $E_{hull}$ as before. Using the oracle's labels, we then compute the percentage of cases in which the condition was properly met (Figure 7). The model is able to generate a material with the correct composition the majority of the time but becomes less reliable as the number of atoms in the chemical formula increases. Space group conditioning is more challenging, as it requires precise control and understanding of 3D structure, but the observed 24% is impressive when considering the 230 possible space groups. Generating stable/unstable structures as a binary task is the most challenging, likely because the training dataset is predominantly stable compounds and stability is defined only in reference to existing compounds. Stability is most easily controlled by modulating sampling hyperparameters.

**Infilling Existing Materials**    In many practical settings, sampling and filtering materials from scratch is unnecessary. Good starting materials are often known, and manufacturing processes are easier to adapt to related compositions than develop completely from scratch by making small edits to their composition–often referred to as *template methods* (Kirklin et al., 2015; Saal et al., 2013). To emulate a typical template method, we construct a lookup table that maps each element to elements that have a similar atom radius when in the same oxidation state (code in Appendix C). We choose an element uniformly at random and swap it with a random element chosen from the table. The resulting structure is then relaxed using M3GNet. To improve this strategy using our fine-tuned LLM, we used the infilling prompt (Section 4) to obtain a distribution over elements (modulated with temperature $\tau$) which we use instead of a uniform distribution over swaps. To evaluate our mutation procedure, we sample 3000 structures randomly from the test set and generate perform one mutation-relaxation step for each, using both uniform and language model-guided sampling. In Figure, 7 we show the

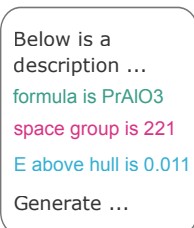 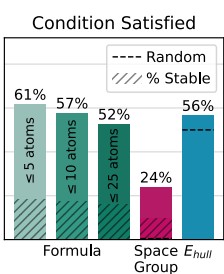 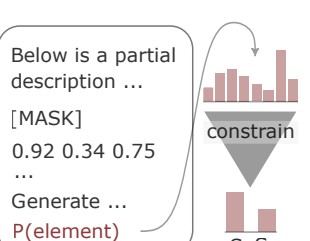 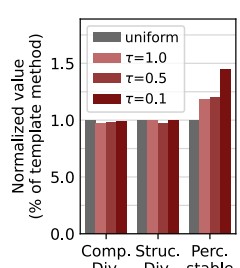

Figure 7: Text-conditional generation and infilling of existing structures with fine-tuned LLMs. (**left**) Including composition or property information (sampled from a hold-out set) in the text prompt leads to a high rate of samples with the desired composition/property (space group or stability). We bin stability as $\hat{E}_{\text{hull}} < 0.1$ (metastable) and $\hat{E}_{\text{hull}} > 0.1$ (unstable) for simplicity. Complex formulas and space groups challenge the model, but the samples are correct at a rate that facilitates practical use. We also show the rate of samples that both satisfy the condition and are predicted to be metastable by M3GNet. (**right**) Using the infilling prompt we can select mutations to existing materials. LLaMA-2 70B proposes a distribution over elements, which we constrain using knowledge of atom radii and charge interactions. We sample mutations with temperature $\tau$ and relax the results structure with M3GNet. When we apply this mutation procedure, we obtain more stable materials per mutation, with negligible changes to the overall diversity of the stable materials.

percentage of stable compounds and diversity in the stable compounds for the uniform baseline and LLaMA-2 70B with different temperature values. LLaMA-2 70B proposes elements that lead to stable structures at a higher rate than the baseline template method without sacrificing diversity.

## 6  DISCUSSION

By generating a high rate of plausible stable materials (verified by DFT), we have demonstrated LLMs can be state-of-the-art generative models for atomistic domains with direct application of parameter-efficient instruction tuning and minimal task-specific modeling choices. This approach to generative modeling opens the door to multitask capabilities within a single sampling paradigm and multimodal training on atoms and text (e.g. to extract knowledge from a large corpus of scientific papers). We also advocate for the use of evaluation metrics (e.g. $E_{\text{hull}}$) for generative models that are more closely tied to the downstream task of generating stable or metastable materials. The space of all hypothetical materials is combinatorially large (consider all the ways to pack 20 arbitrary elements into a box), but only a small subset of materials will actually be stable or metastable. Models that can directly generate near-stable structures make all downstream tasks far easier, and increases the likelihood the generative models may be useful for day-to-day tasks in materials discovery.

**Limitations**  Our method shares the limitations of the underlying generative models. LLMs can be sensitive to precise details of the chosen prompt and the tokenization strategies, particularly in how tokenization effects processing of numbers. Hallucination of unphysical chemical elements or structures has been observed, though fortunately is easy to check and filter. Text-conditioning has the potential to tap latent conceptual understanding in the underlying LLM, but training LLMs that successfully leverage scientific and chemistry literature is a major outstanding challenge. Lastly, training the largest of our LLMs can be prohibitively expensive for some computational budgets. Despite this, inference from all LLMs is often highly tractable when compared to baseline methods (Appendix B.7).

**Future directions**  There is substantial room for improvement in conditional generation, which could be used to directly generate materials with desired properties. While we did not pursue alternative sampling strategies in depth, approaches like classifier-free guidance (Sanchez et al., 2023) or variants of PPLM (Dathathri et al., 2019) might be useful in combination with fine-tuned LLMs to improve conditional generation. These methods could also be combined with primitives from Bayesian optimization for sample-efficient and uncertainty-aware design (Stanton et al., 2022; Gruver et al., 2023b).

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

# Appendix

## Table of Contents

## A  TRAINING DETAILS

### A.1  NUMERICAL FORMATTING

Notably, our approach to tokenization is distinctly different from prior work on modeling atomic structures with language models. Instead of using a special vocabulary and training models from scratch, we use LLaMA-2's existing tokenizer. This choice allows us to easily process both encoded crystals and text data. In early experiments, we tried out many other approaches, including fine-tuning LLaMA-2 models with additional tokens specific to crystal data. These methods were more challenging to train and didn't lead to any improvements over using a shared tokenizer. We include a set of example training losses below:

| | Epoch 1 | Epoch 2 | Epoch 3 | Epoch 4 | Epoch 5 |
|---|---|---|---|---|---|
| Special Crystal Tokens | 0.783 | 0.693 | 0.623 | 0.611 | 0.588 |
| Shared Tokenization | 0.457 | 0.432 | 0.424 | 0.401 | 0.385 |

There are many important decisions involved both in text formatting (e.g the choice of fractional or absolute coordinates) and augmentation of the input data (e.g. translation or permutation augmentations on coordinates). As a simple example, we provide average validity numbers (using low temperature sampling) from earlier experiments on LLaMA-2 7B models trained with different formatting styles

| Setting | Structural Validity | Compositional Validity |
|---|---|---|
| Fractional coords | 91.4% | 83.2% |
| Absolute coords | 90.8% | 80.5% |
| No permutations | 92.5% | 82.9% |
| With permutations | 89.2% | 81.7% |

### A.2  TRAINING WITH STOCHASTIC PROMPTS

In order to enable multi-task use of the fine-tuned LLMs, we train on a stochastically generated prompt. Two thirds of the time we provide the model with a generation task, in which the prompt

consists of a basic instruction to generate a bulk material as a lattice and atom positions. We randomly sample a set of properties from the available descriptors of a given crystal and add any chosen ones (if any) to the prompt, using a small amount of wrapper text. The remaining one third of the time, we provide use the sampled crystal to construct and infilling task. We choose on element randomly from the set of elements in the composition and we construct a prompt that contain the string encoding of the crystal with this element replaced with [MASK]. The model then generates the replaced element as text following the prompt.

## A.3 EXTENDED MATERIALS PROJECT DATASET

To facilitate text-conditional generation, we extend the original CDVAE training dataset with materials from Materials Project (Jain et al., 2013) as of April 2023. We filter out crystal with more than 30 atoms in the unit cell, which slow down training with minimal benefit to model performance, leaving a training set that contains 127609 crystal structures. The original validation and test splits are left unchanged and all test/validation points are removed from the new training set.

## A.4 TRAINING HYPERPARAMETERS AND DETAILS

We provide the training details per model:

- **LLaMA-2 7B**: Batch size of 256 for 65 epochs with a cosine annealed learning rate of 0.0005. LoRA rank 8 and alpha 32.

- **LLaMA-2 13B**: Batch size of 256 for 44 epochs with a cosine annealed learning rate of 0.0005. LoRA rank 8 and alpha 32.

- **LLaMA-2 70B**: Batch size of 32 for 21 epochs with a cosine annealed learning rate of 0.0005. LoRA rank 8 and alpha 32.

Limitations around available compute lead to our use of differing batch sizes and total number of epochs for each model. Ideally, we would train all models with the largest batch sized used among all models and would train all models for the same number of epochs (the maximum used by any model). At the same time, we wanted to properly demonstrate the full potential of all model sizes and therefore chose to present results for the best model we were able to train at each model size.

## A.5 ROLE OF TEXT PRETRAINING

Text pretraining is essential to our method for two reasons.

1. It would be impractically expensive or computationally infeasible to train models with up to 70B parameters from scratch on our data. Using a pretrained model with LoRA (Hu et al., 2021) offers the benefits of model scale while maintaining tractability and limiting overfitting, as the actual number of trainable parameters can be relatively small.

2. Pretraining on text data yields a model that can be conditioned on text for free, and text conditioning opens up a huge new realm of exciting possibilities, like conditioning samples on desired properties. It would be challenging to achieve a similar result from scratch without significantly expanding the size of the dataset (to improve general text understanding) and without essentially training a general-purpose language model in the process.

To better understand the first point, let's quickly review the exact details of the finetuning procedure. We are using low-rank adapters (LoRA), as opposed to end-to-end finetuning, and this means we are adding a small number of additional parameters to an existing, frozen model. The easiest way to see the difference between this approach and training a model from scratch–as in (Flam-Shepherd & Aspuru-Guzik, 2023)–is to compare the training loss over the first few epochs of training.

| Model | Epoch 1 | Epoch 2 | Epoch 3 | Epoch 4 | Epoch 5 |
|---|---|---|---|---|---|
| GPT-2 (from scratch) | 0.946 | 0.878 | 0.807 | 0.757 | 0.740 |
| LLaMA-13B (LoRA) | 0.457 | 0.432 | 0.424 | 0.401 | 0.385 |
| LLaMA-70B (LoRA) | 0.402 | 0.344 | 0.325 | 0.305 | 0.296 |

If we attempt to run LoRA finetuning with randomly initialized parameters for the LLaMA-2 7B model we observe an immediate and significant difference in the training losses:

| Model | 1 Iter | 0.33 Epochs | 0.66 Epochs | 1 Epoch |
|---|---|---|---|---|
| Random | 13.46 | 1.53 | 0.81 | 0.78 |
| Pre-trained | 1.57 | 0.47 | 0.41 | 0.39 |

While LoRA finetuning is tractable because 99.95% of the model is frozen, finetuning a LLaMA-2 model end-to-end in half-precision would require at least 4 times as many GPUs, making it infeasible for all but a handful of researchers. When using LoRA, even though the base models are large the number of trainable parameters is very small. In fact, the LLamA-2 7B model has less trainable parameters than one of the baseline methods we compared (CDVAE) (Xie et al., 2021). The number of trainable parameters for each of our models and the baseline models is shown below:

| Model | Trainable parameters (millions) | Percentage of total |
|---|---|---|
| CDVAE | 4.5 | 100% |
| LM-CH/AC | 1-100 | 100% |
| LLaMA-2 7B | 3.5 | 0.05% |
| LLaMA-2 13B | 6.5 | 0.05% |
| LLaMA-2 70B | 35 | 0.05% |

# B  MODEL EVALUATION

## B.1  EVALUATION WITH ML POTENTIALS AND DFT

Approximating $E_{\text{hull}}$ from the energies of known materials in Materials Project requires a consistent correction scheme. We touch on some of the details here.

**M3GNet**  Importantly, M3GNet was trained on the total energy of VASP calculations in the Materials Project dataset, so the results were expected to be consistent with the correction schemes and absolute energies in Section 5.

**VASP**  To be consistent with the Materials Project settings (e.g. the PBE functional, DFT/DFT+U as appropriate, consistent pseudopotentials, etc). We did a single relaxation for every candidate structure using the default parameters in MPRelaxSet (Ong et al., 2013). VASP relaxations were run using the GPU-accelerated VASP6 code.

In both situations, the total energies were corrected using the MP2020 compatibility scheme, which was important to maintain consistency when calculating formation energies, and allow the use of varying functionals (DFT/DFT+U) for different materials.

## B.2  STABILITY CHECKS AND PERCENTAGES

To calculate the percentage of metastable compounds, we take all samples and remove samples that are invalid under the basic structure and composition checks. We then run relaxations with M3GNet and obtain the final relaxation energies. The final percentage takes into account both the rate of validity (used to perform the initial filtering), and the rate of compounds with $\hat{E}_{\text{hull}} < 0.1$, as determined by the convex hull calculation using the M3GNet relaxation energy. To calculate the VASP percentage, we select materials determined to be metastable M3GNet and run VASP with default setting. We then report the percentage of the materials with $\hat{E}_{\text{hull}} < 0.0$.

## B.3  TRADE-OFFS IN SAMPLING

We note that modulating stability with sampling parameters like temperature and nucleus size has a significant effect on the coverage properties of the resulting samples. We illustrate the trade-offs between stability and coverage in Figure 8. Coverage most likely decreases because nucleus size and temperature collapse the distribution around samples with high likelihood, which are also more likely to be valid or stable. Notably, LLaMA-2 70B appears to demonstrate the best trade-offs, possibly

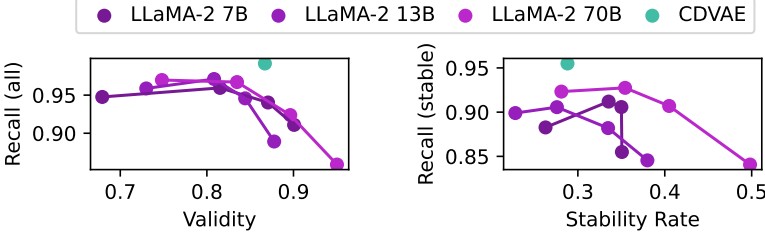

Figure 8: Validity and rate of stability depend on sampling hyper-parameters. Lowering the temperature or restricting the nucleus size leads to significant improvements in validity/stability but incurs a cost to coverage of a held-out test set (recall). Fine-tuned LLaMA-2 70B displays the best trade-off between coverage and stability, generating materials that are both stable and diverse.

indicating a likelihood model that corresponds better to both the underlying properties of stability and the full, diverse distribution of structures.

### B.4 "HALLUCINATION" EXAMPLES

**LLaMA-2 7B**:

```
# generated using pymatgen
data_Met8(Cu2N)5
_symmetry_space_group_name_H-M    'P 1'
_cell_length_a      5.0000
_cell_length_b      5.0000
_cell_length_c      5.0000
_cell_angle_alpha      90.0000
_cell_angle_beta      90.0000
_cell_angle_gamma      90.0000
_symmetry_Int_Tables_number      1
_chemical_formula_structural      Met8(Cu2N)5
_chemical_formula_sum      'Met8 Cu10 N5'
_cell_volume      125.0000
_cell_formula_units_Z      1
loop_
 _symmetry_equiv_pos_site_id
 _symmetry_equiv_pos_as_xyz
  1   'x, y, z'
loop_
 _atom_site_type_symbol
 _atom_site_label
 _atom_site_symmetry_multiplicity
 _atom_site_fract_x
 _atom_site_fract_y
 _atom_site_fract_z
 _atom_site_occupancy
  Cu   Cu0   1   1.8300   0.3900   1.0000   1
  Cu   Cu1   1   0.8300   0.4900   1.0000   1
  Cu   Cu2   1   0.8300   0.9900   0.5000   1
  Cu   Cu3   1   0.6300   0.1900   0.2000   1
  Cu   Cu4   1   0.2300   0.7900   0.2000   1
  Cu   Cu5   1   0.6300   0.7000   0.3100   1
  Cu   Cu6   1   0.2300   0.1900   0.3000   1
  Cu   Cu7   1   1.0000   0.8900   0.7000   1
  Cu   Cu8   1   1.0000   0.3900   0.2000   1
  Cu   Cu9   1   0.4900   0.8900   0.7000   1
  Met0+   Met10   1   0.6300   0.6000   1.0000   1
  Met0+   Met11   1   0.4000   0.4700   0.4700   1
  Met0+   Met12   1   0.4000   1.0000   0.9800   1
  Met0+   Met13   1   1.0000   0.2200   0.9700   1
  Met0+   Met14   1   1.0000   0.6300   0.5000   1
  Met0+   Met15   1   0.2300   0.2200   0.6000   1
  Met0+   Met16   1   1.0000   0.0000   0.6100   1
  Met0+   Met17   1   0.6300   0.1000   0.5000   1
  N   N18   1   0.1200   0.7000   0.8000   1
  N   N19   1   0.2300   0.5900   0.2000   1
  N   N20   1   0.2300   0.1900   0.7000   1
  N   N21   1   0.4900   0.2100   0.1000   1
  N   N22   1   0.4800   0.6100   0.6000   1
```

```
data_L3Li
_symmetry_space_group_name_H-M    'P 1'
_cell_length_a      5.1000
_cell_length_b      7.1000
_cell_length_c      7.4000
_cell_angle_alpha      84.0000
_cell_angle_beta      68.0000
_cell_angle_gamma      68.0000
_symmetry_Int_Tables_number      1
_chemical_formula_structural      L3Li
_chemical_formula_sum      'L12 Li4'
_cell_volume      230.15214369
_cell_formula_units_Z      4
loop_
 _symmetry_equiv_pos_site_id
 _symmetry_equiv_pos_as_xyz
  1   'x, y, z'
loop_
 _atom_site_type_symbol
 _atom_site_label
 _atom_site_symmetry_multiplicity
 _atom_site_fract_x
 _atom_site_fract_y
 _atom_site_fract_z
 _atom_site_occupancy
  Li   Li0   1   0.7100   0.4000   0.8300   1
  Li   Li1   1   0.2200   0.3700   0.3600   1
  Li   Li2   1   0.7100   0.8900   0.3300   1
  Li   Li3   1   0.2100   0.8700   0.8600   1
  L0+   L4   1   1.0000   0.6300   0.6900   1
  L0+   L5   1   0.5100   0.1400   0.6600   1
  L0+   L6   1   0.9600   0.5700   0.1700   1
  L0+   L7   1   0.4700   0.0700   0.1700   1
  L0+   L8   1   0.9800   0.6100   0.1400   1
  L0+   L9   1   0.4900   0.1100   0.1400   1
  L0+   L10   1   1.0000   0.1000   0.6800   1
  L0+   L11   1   1.0000   0.1100   0.1500   1
  L0+   L12   1   0.4700   0.5500   0.1800   1
  L0+   L13   1   1.0000   0.5800   0.6800   1
  L0+   L14   1   0.4700   0.0600   0.6700   1
  L0+   L15   1   1.0000   0.1300   0.1700   1
```

**LLaMA-2 13B**:

```
data_LeB7(NO3)2
_symmetry_space_group_name_H-M    'P 1'
_cell_length_a     5.9000
_cell_length_b     5.9000
_cell_length_c     5.9000
_cell_angle_alpha      59.0000
_cell_angle_beta       59.0000
_cell_angle_gamma      59.0000
_symmetry_Int_Tables_number     1
_chemical_formula_structural      LeB7(NO3)2
_chemical_formula_sum      'Le1 B7 N2 O6'
_cell_volume    141.91223582
_cell_formula_units_Z    1
loop_
 _symmetry_equiv_pos_site_id
 _symmetry_equiv_pos_as_xyz
  1   'x, y, z'
loop_
 _atom_site_type_symbol
 _atom_site_label
 _atom_site_symmetry_multiplicity
 _atom_site_fract_x
 _atom_site_fract_y
 _atom_site_fract_z
 _atom_site_occupancy
  Le0+  Le0  1   0.7100   0.5000   0.1700   1
  B    B1   1   0.3800   0.1600   0.0200   1
  B    B2   1   0.4600   0.1600   0.5700   1
  B    B3   1   0.4600   0.7200   0.5700   1
  B    B4   1   0.0400   0.7900   0.6500   1
  B    B5   1   1.0000   0.2500   0.6500   1
  B    B6   1   0.0000   0.7900   0.0900   1
  B    B7   1   0.0000   0.1600   0.6500   1
  N    N8   1   0.6200   0.5700   0.9800   1
  N    N9   1   0.0600   0.3300   0.2500   1
  O    O10  1   0.5500   0.7600   0.7100   1
  O    O11  1   0.1800   0.5400   0.6100   1
  O    O12  1   0.4300   0.9500   0.5400   1
  O    O13  1   0.9400   0.1100   0.9600   1
  O    O14  1   0.6400   0.7700   0.2900   1
  O    O15  1   0.3000   0.3800   0.1300   1
```

```
data_MandeGd2O4
_symmetry_space_group_name_H-M    'P 1'
_cell_length_a     3.6000
_cell_length_b     3.6000
_cell_length_c     5.9000
_cell_angle_alpha      90.0000
_cell_angle_beta       90.0000
_cell_angle_gamma      90.0000
_symmetry_Int_Tables_number     1
_chemical_formula_structural      MandeGd2O4
_chemical_formula_sum      'Mande1 Gd2 O4'
_cell_volume    76.46400000
_cell_formula_units_Z    1
loop_
 _symmetry_equiv_pos_site_id
 _symmetry_equiv_pos_as_xyz
  1   'x, y, z'
loop_
 _atom_site_type_symbol
 _atom_site_label
 _atom_site_symmetry_multiplicity
 _atom_site_fract_x
 _atom_site_fract_y
 _atom_site_fract_z
 _atom_site_occupancy
  Gd    Gd0    1   0.8200   0.2300   0.1500   1
  Gd    Gd1    1   0.8200   0.2300   0.6300   1
  Mande0+  Mande2  1   0.3200   0.7300   0.8900   1
  O    O3    1   0.8200   0.7300   0.4100   1
  O    O4    1   0.3200   0.7300   0.1000   1
  O    O5    1   0.3200   0.2300   0.3900   1
  O    O6    1   0.8200   0.7300   0.7900   1
```

**LLaMA-2 70B**:

```
data_Ln3BO4
_symmetry_space_group_name_H-M    'P 1'
_cell_length_a     5.3000
_cell_length_b     5.9000
_cell_length_c     5.3000
_cell_angle_alpha      62.0000
_cell_angle_beta       90.0000
_cell_angle_gamma      90.0000
_symmetry_Int_Tables_number     1
_chemical_formula_structural      Ln3BO4
_chemical_formula_sum      'Ln3 B1 O4'
_cell_volume    146.33178751
_cell_formula_units_Z    1
loop_
 _symmetry_equiv_pos_site_id
 _symmetry_equiv_pos_as_xyz
  1   'x, y, z'
loop_
 _atom_site_type_symbol
 _atom_site_label
 _atom_site_symmetry_multiplicity
 _atom_site_fract_x
 _atom_site_fract_y
 _atom_site_fract_z
 _atom_site_occupancy
  Ln0+  Ln0  1   0.1800   0.0600   0.9900   1
  Ln0+  Ln1  1   0.6800   0.5600   0.9900   1
  Ln0+  Ln2  1   0.1800   0.5600   0.4900   1
  B    B3   1   0.6800   0.0600   0.4900   1
  O    O4   1   0.6800   0.3300   0.1500   1
  O    O5   1   0.1800   0.2800   0.1800   1
  O    O6   1   0.6800   0.7800   0.8000   1
  O    O7   1   0.1800   0.8300   0.8500   1
```

```
data_Gro15Nd4
_symmetry_space_group_name_H-M    'P 1'
_cell_length_a     7.0000
_cell_length_b     7.0000
_cell_length_c     6.9000
_cell_angle_alpha      71.0000
_cell_angle_beta       71.0000
_cell_angle_gamma      69.0000
_symmetry_Int_Tables_number     1
_chemical_formula_structural      Gro15Nd4
_chemical_formula_sum      'Gro15 Nd4'
_cell_volume    289.96945358
_cell_formula_units_Z    1
loop_
 _symmetry_equiv_pos_site_id
 _symmetry_equiv_pos_as_xyz
  1   'x, y, z'
loop_
 _atom_site_type_symbol
 _atom_site_label
 _atom_site_symmetry_multiplicity
 _atom_site_fract_x
 _atom_site_fract_y
 _atom_site_fract_z
 _atom_site_occupancy
  Nd    Nd0   1   0.5600   0.5700   0.7800   1
  Nd    Nd1   1   0.7500   0.7500   0.5600   1
  Nd    Nd2   1   0.1700   0.1700   0.1400   1
  Nd    Nd3   1   0.9500   0.9500   0.3800   1
  Gro0+  Gro4   1   0.7600   0.2300   0.3000   1
  Gro0+  Gro5   1   0.1200   0.4800   1.0000   1
  Gro0+  Gro6   1   0.3800   0.8700   0.1000   1
  Gro0+  Gro7   1   0.0300   0.6600   0.8400   1
  Gro0+  Gro8   1   0.6500   0.1700   0.6400   1
  Gro0+  Gro9   1   0.5600   0.0600   0.7400   1
  Gro0+  Gro10  1   0.9200   0.5000   0.1600   1
  Gro0+  Gro11  1   0.4900   0.7400   0.2200   1
  Gro0+  Gro12  1   0.2400   0.1000   0.5800   1
  Gro0+  Gro13  1   0.9100   0.2700   0.6200   1
  Gro0+  Gro14  1   0.4000   0.6100   0.4600   1
  Gro0+  Gro15  1   0.2900   0.2900   0.4200   1
  Gro0+  Gro16  1   0.4500   0.9200   0.9400   1
  Gro0+  Gro17  1   0.9900   0.1300   0.0200   1
  Gro0+  Gro18  1   0.8400   0.5100   0.8200   1
```

### B.5    Increase in Perplexity under Transformation (IPT)

Although there are existing metrics for invariance and equivariance in neural networks, language models pose unique challenges because of their discrete tokens, which do not change smoothly under continuous transformations. Though it might be possible to compute a meaningful analogue of the Lie derivative (Gruver et al., 2022), or similar metrics, through interpolation of word embeddings, we decide to adopt a simpler metric (IPT), which still highlights significant differences between base models. We calculate IPT for each model using 500 test datapoints and 20 randomly translation sampled as fraction coordinates from a uniform distribution per dimension. The translations themselves are implemented in PyMatgen and respect periodic boundary conditions (Ong et al., 2013). In order to combine the IPT values in a meaningful way across different datapoints, we normalize their values by the mean perplexity over transformations. Thus datapoints which happen to have large perplexity, and therefore naturally large potential changes in perplexity, do not drown out points with small perplexity.

### B.6    Diversity and Novelty Calculation

Following (Xie et al., 2021), we calculate diversity as the pairwise distance between samples using a featurization of structure and composition. To calculate novelty, we also featurize the training dataset and calculate the distance to the nearest element of the training set for each sample. A sample is considered novel if the nearest element in the training set is above a threshold. We use a structural distance cutoff of 0.1 and composition distance cutoff of 2. In addition to novelty of structure and composition individual, we also consider the overall novelty of a crystal, where overall novelty is determined by having either a new structure or a new composition. All metrics are calculated on filtered samples that M3GNet qualifies as metastable. We report metrics on metastable samples because these numbers are more practically relevant and because the samples are more likely to contribute meaningful variation, instead of being different from the training set and each other simply because they are wildly invalid. We normalize all diversity and novelty values by corresponding value for the test set to provide a sense for the underlying data distribution.

### B.7    Sampling Speed

Although LLMs might seem like computational overkill at face value, batching for large-scale sampling allows LLaMA models to have comparable computational overhead to competing approaches. Making exact comparisons between LLaMA models and CDVAE are slightly challenging because of available hardware and differences in compatibility. We ran experiments primarily on A100 GPUs, while the publicly available code for CDVAE cannot be run on an A100 and reports results on a RTX2080 Ti.

We provide two analyses for the sampling rate of LLaMA models, one from experiments we ran on a single A100 and alternative using third-party numbers for LLaMA models deployed on AWS instances.

**Local analysis**    We obtain benchmark LLaMA-2 sampling times by running 5 batched generations and computingn the average time to completion. We then use these numbers to calculate the equivalent time to sample 10,000 structures. In practice, we used distributed sampling on a cluster, so reporting our direct times to compute 10,000 samples would be less informative. We use the maximum batch size that we can fit on an A100 GPU with each model without causing out-of-memory (OOM) errors during sampling. The batch sizes were {7B: 512, 13B: 256, 70B: 128}. To compare CDVAE with our results we perform a rough, but generous, conversion of their results to an A100 GPU. We multiply their rate of sampling by 16, to account for the 2x faster rate of operations (Balaban, 2020) and approximately 8 times larger GPU memory (allowing for large batch sizes and utilization rates). We report the intermediate numbers and calculations below. The final rates for metastable samples are shown in Figure 6.

| Model | Batch size | Seconds / batch | Samples / hour | Hours / 10,000 crystals |
|---|---|---|---|---|
| CDVAE | 512 | n/a | n/a | 1.260 |
| LLaMA-2 7B | 512 | 27.18 | 67814 | 0.147 |
| LLaMA-2 13B | 256 | 38.24 | 24100 | 0.414 |
| LLaMA-2 70B | 128 | 52.52 | 8774 | 1.139 |

**AWS analysis** Considering AWS as the deployment environment, we can build on a recent benchmark on a cloud instance with 8 A100 GPUs (ml.p4d.12xlarge) (Schmid, 2023), which found that LLaMA-2 13B achieved 0.416 hr/1M tokens and LLaMA-2 70B achieved 0.864 hr/1M tokens. One crystal is around 100 tokens on average, so the throughput for 10,000 crystals is the same as for 1M tokens. For comparison, we use CDVAE and its recorded runtimes for generating 10,000 crystals on a single RTX2080 Ti GPU (Xie et al., 2021). To obtain the final numbers, we adjust for the number of GPUs (8) and a 2x improvement from RTX2080 Ti to A100 GPUs (Balaban, 2020).

| Model | Hours / 10,000 crystals | Hours / 10,000 metastable (M3GNet) crystals |
|---|---|---|
| CDVAE | 0.363 | 1.260 |
| LLaMA-2 13B | 0.416 | 1.094 |
| LLaMA-2 70B | 0.864 | 1.728 |

We see that LLaMA-2 13B actually has a comparable computational overhead to prior work, and LLaMA-2 70B is only slightly higher. When considering the rate of stable materials generated by each method, we see that LLaMA-2 13B actually has a higher throughput than CDVAE.

## C   TEMPLATE METHOD BASELINE

We provide code in Listing 1 implementing construction of the physically-inspired element swap table. This table is used by both the template method and the LLM-guided sampling method to constrain search to elements that are physically plausible. Listing 2 shows our implementation of a basic template method with uniform sampling. The LLM-guided procedure is mostly identical, except with uniform sampling of the swap element changed for sampling from a distribution obtained from the LLM with an infilling prompt (and modulated with temperature parameter $\tau$)

```
1  import os
2  import random
3  import pandas as pd
4  import numpy as np
5  from pymatgen.core import Element
6  from pymatgen.core.structure import Structure
7  from m3gnet.models import Relaxer
8
9  def find_similar_elements(target_element, elements, tolerance=0.1):
10     similar_elements = []
11     for state, radius in target_element.ionic_radii.items():
12         for el in elements:
13             if state in el.ionic_radii:
14                 radius_diff = abs(radius - el.ionic_radii[state])
15                 if radius_diff < tolerance and el.symbol !=
    target_element.symbol:
16                     similar_elements.append((el.symbol, state,
    radius_diff))
17     return sorted(similar_elements, key=lambda x: x[2])
18
19  def make_swap_table():
20     elements = [Element(el) for el in Element]
21
22     swap_table = {}
23
24     for el in elements:
25         swap_table[el.symbol] = [
26             x[0] for x in find_similar_elements(el, elements)
27         ]
```

```
28
29     return swap_table
```

Listing 1: Self contained code to construct the template method table which can be used to proposed mutations for local optimization around an existing material. The same table can be used in tandem with a language model to provide sampling constraints (i.e. eliminate elements which are very physically unlikely).

```
1  def propose_new_structures(cif_str, swap_table, max_swaps=1):
2      struct = Structure.from_str(cif_str, fmt="cif")
3
4      elements = [el.symbol for el in struct.species]
5      swappable_elements = [
6          el for el in elements if el in swap_table and len(swap_table[el])
        > 0
7      ]
8
9      num_possible_swaps = sum([len(swap_table[el]) for el in
       swappable_elements])
10     num_swaps = min(num_possible_swaps, max_swaps)
11
12     relaxer = Relaxer()
13     new_bulks = []
14     for _ in range(num_swaps):
15         old_el = random.choice(swappable_elements)
16         possible_new = swap_table[old_el]
17         new_el = random.choice(possible_new)
18
19         new_bulk = struct.copy()
20         new_bulk.replace_species({old_el: new_el})
21
22         relax_results = relaxer.relax(new_bulk)
23         final_structure = relax_results['final_structure']
24         final_relaxed_energy = relax_results['trajectory'].energies[-1]
25
26         new_bulks.append(dict(
27             cif=final_structure.to(fmt="cif"),
28             energy=final_relaxed_energy
29         ))
30
31     new_bulks = pd.DataFrame(new_bulks)
32     return new_bulks
```

Listing 2: Self contained code implementing a template method with uniform sampling. Our language model procedure is essentially the same but replaces uniform sampling with logits from a prompted language model. This language model can use the context from the rest of the crystal structure to propose a mutation instead of choosing a mutation completely at random.

