# OpenReview forum: "Fine-Tuned Language Models Generate Stable Inorganic Materials as Text"
_ICLR.cc/2024/Conference — ICLR 2024 poster_

### Official Review · Reviewer_1Fky · 2023-10-29

**Soundness:** 2 fair
**Presentation:** 2 fair
**Contribution:** 2 fair
**Rating:** 6
**Confidence:** 5

**Summary:**

The paper proposes an approach to materials discovery by fine-tuning large language models (LLMs) on text-encoded atomistic data. The authors claim that this method can generate materials predicted to be metastable at a higher rate than competing diffusion models.

**Strengths:**

* The paper is well-motivated, addressing the limitations of existing computational materials databases and the potential of generative models for materials discovery.
* The proposed approach of fine-tuning large language models on text-encoded atomistic data is novel and unorthodox, offering a new perspective on materials generation.
* The paper is well-written, providing clear background information and a thorough explanation of the proposed method.

**Weaknesses:**

Related concerns are discussed in the questions section.

**Questions:**

* Can the authors provide a more detailed comparison with existing generative models in materials discovery, discussing the advantages and limitations of the proposed LLM-based approach compared to other state-of-the-art methods?
* Are there any potential drawbacks or limitations of using LLMs for materials discovery, such as computational complexity or interpretability of the generated structures?
* In the paper, the authors mention that "We chose not to augment the ordering of atoms because these variables often contained valuable information". How to ensure that the atomic order will not change the results?
* In the paper, the authors mention that "we only run VASP calculations on materials that have already been predicted as metastable by M3GNet", while in the caption of figure 3, "we only run VASP on structures predicted to be stable by M3GNet." What are the differences between the "stable" and "metastable"?

---

> ### Author Response · Authors · 2023-11-20
> **Author Rebuttal to Reviewer 1Fky**
>
> Thank you for your review and your supportive comments. We respond to your questions pointwise. Note we have a separate [general post](https://openreview.net/forum?id=vN9fpfqoP1&noteId=wbNkvYNin2) outlining the key contributions in the paper.
>
> **Existing generative models**
>
> State-of-the-art generative models of materials from recent work can be grouped in two categories (1) diffusion models trained to denoise atomic coordinates (2) language models trained from scratch on crystal data.
>
> **(1) Overview**
>
> - *Diffusion models*: Methods like CDVAE [1] and DiffCSP [2] learn a distribution over crystal structures by learning a neural network that takes noise-corrupted coordinates and/or lattice parameters and predicts the original input without noise. During sampling, examples are generated from noise through repeated applications of the denoising model. Because these methods operate on continuous representations of the coordinates and lattice (as opposed to discrete string representation), they can naturally incorporate symmetries into the architecture of the denoising model, for example translational equivariance.
>
> - *Language models*: Prior work on language models for crystals, e.g. [3,4], focuses on training models from scratch on crystal data. In these papers, GPT-2 style models with 10-100 million parameters are trained on custom vocabularies to represent element identities and atomic coordinates. [3] uses a relatively small dataset of around 50,000 materials while [4] uses millions of CIF files. [3] applies augmentations to encourage invariance in the model while [4] does not.
>
> - *Our method*: We finetune a pretrained large language model using a relatively small number of additional trainable parameters (between 3 and 35 million). We use LLaMA-2’s default tokenizer, which allows us to train on both crystal data and English text prompts. We apply augmentations to encourage invariance and measure learned invariance with a new metric useful to many applications of LLMs on atomistic data.
>
> **(2) Capabilities**
>
> - *Diffusion models*: Because diffusion models are trained to correct noisy inputs, they are often directly applicable to infilling tasks, e.g. for editing a small part of an existing structure. Additionally, because diffusion models act directly on continuous representations of coordinates they often exhibit higher rates of structural validity because notions of distance between atoms are more easily accessible to the model. On the other hand, compared to autoregressive language models, diffusion models often display lower rates of compositional validity, likely because compositions are naturally discrete sequences, and autoregressive language models remain state of the art generative models for discrete sequence.
>
> - *Language models*: Because language models are commonly used for text processing, they naturally incorporate text-conditional generation, which is explored to a limited degree in [4] but not in [3]. By default, language models are slightly less natural for editing tasks, unless trained with carefully chosen prompts, which is not explored in prior work. Because language models operate on string representations of atomic coordinates, they often have lower rates of structural validity because notions of distance/overlapping atoms must be learned over discrete sequences.
>
> - *Our method*: By using pretrained language models and carefully engineered prompts, we can achieve the best of both worlds. Our models are capable of editing existing structures and conditioning on complex English prompts. Text pretraining also leads to models that more easily learn programmatic abstractions over text, e.g. for analyzing positions in 3D space. Thus we see that finetuned LLMs, especially larger models like LLaMA-2 70B, achieve high rates of both structural and compositional validity.
>
> **(3) Generality**
>
> - *Diffusion models*: Models like CDVAE [1] and DiffCSP [2] are tailored specifically to generating crystals. The same methodology can not be applied to other atomistic data without major changes to the architecture and training procedure.
>
> - *Language models*: Unlike the diffusion models above, the training procedures used for language modeling are easier to generalize across different forms of atomistic data, as shown in [3]. When using custom vocabularies, however, the model weights themselves cannot be shared, despite shared training and sampling procedures.
>
> - *Our method*: By using a general-purpose tokenization method and a pretrained language model, we create a procedure that applies equally to natural language and string-formatted atomistic data. Though we focused primarily on crystals, a single LLM could also be trained simultaneously on other data types, such as small organic molecules or proteins.

---

> > ### Author Response · Authors · 2023-11-20
> > **Author Rebuttal to Reviewer 1Fky (part 2)**
> >
> > **(4) Results**
> >
> > Our method achieves high levels of validity and stability, as shown in the condensed table below:
> >
> > | Model | Struct. Validity | Comp. Validity | Metastability (M3GNet) | Stability (DFT) |
> > | -------- | -------- | -------- | -------- | -------- |
> > | CDVAE | 100% | 86.7% | 28.8% | 5.4% |
> > | LM-CH | 84.8% | 83.5% | n/a | n/a |
> > | LM-AC | 95.8% | 88.9% | n/a | n/a |
> > | LLaMA-2 70B ($\tau$ = 1.0) | 96.5% | 86.3% | 35.4% | 10.0% |
> > | LLaMA-2 70B ($\tau$ = 0.7) | 99.6% | 95.4% | 49.8% | 10.6% |
> >
> > **(5) Computational cost**
> >
> > As we lay out in the general rebuttal note, we can calculate the following sample throughputs for a standard deployment environment on AWS:
> >
> > | Model | Hours / 10,000 crystals | Hours / 10,000 metastable (M3GNet) crystals |
> > | -------- | -------- | -------- |
> > | CDVAE | 0.363 | 1.260 |
> > | LLaMA-13B | 0.416 | 1.094 |
> > | LLaMA-70B | 0.864 | 1.728 |
> >
> > **(6) Conclusion**
> >
> > From this comparison it is evident our method has significantly expanded capabilities and generality while achieving strong performance and only slightly greater computational cost.

---

> > > ### Author Response · Authors · 2023-11-20
> > > **Author Rebuttal to Reviewer 1Fky (part 3)**
> > >
> > > **Limitations of LLMs**
> > >
> > > Currently LLMs are relatively compute expensive, although by a smaller margin than one might expect. As we outline in the general rebuttal note, smaller LLMs (13B) have throughput on par with many baseline models, while larger models (70B) are only 2-3 times slower in throughput. Recently released models, e.g. Mistral 7B [5], have also matched the performance of large models with significantly less parameters. We expect this general trend to continue and lead to faster sampling times, as the demand for tractable base models has only intensified.
> > >
> > > Current conditional generation from LLMs is not perfect. There are cases in which the composition is specified but the resulting crystal is inconsistent with this composition. Strict instruction following can be an issue for LLMs in general, and authors have proposed approaches that we could also apply in our setting, such as classifier-free guidance [6]. If we wanted to strictly enforce composition constraints, we could adapt the infilling framework to generate positions while keeping atom identities fixed, but we chose to use the more general framework for simplicity and because vanilla composition conditioning was successful frequently enough to be practical.
> > >
> > > Hallucinations of non-existent elements can occur when using vanilla sampling from the model. If these mistakes are left unhandled, they can waste compute on samples that have no chance of being valid. Luckily, it is easy to prevent these mistakes with simple constrained sampling techniques. In the simplest case, we can simply lower the likelihood of tokens that do not occur in atom identities to zero. For ambiguous tokens that can occur within element identities or other non-element strings, we can apply sampling utilities available in huggingface [7].
> > >
> > > **Role of atomic ordering**
> > >
> > > Although the global ordering of the atoms is arbitrary, the ordering in the data typically contains meaningful structure. For example, consider the following crystal for Pd3(PbSe)2, taken directly from Materials Project [8],
> > >
> > > ```
> > > Lattice
> > >     abc: 6.29 6.29 6.34
> > >     angles: 60.3 60.3 60.0
> > > Coordinates (fractional)
> > > 	Pd (0.5, 0.5, 0.0)
> > > 	Pd (0.5, 0.0, 0.0)
> > > 	Pd (0.0, 0.5, 0.0)
> > > 	Pb (0.5, 0.5, 0.5)
> > > 	Pb (0.0, 0.0, 0.0)
> > > 	Se (0.21, 0.21, 0.36)
> > > 	Se (0.79, 0.79, 0.64)
> > > ```
> > >
> > > The elements are grouped together, and there are patterns evident in the atom placements within each element type. For example, we can imagine placing the Palladium atoms first, then the lead atoms at opposite corners in a cubic structure, and lastly the Selenium atoms can be placed between the existing atoms. In earlier versions of our method, we experimented with permuting the order of the atoms, and the model was still able to learn effectively, but the sample metrics (e.g. validity) were worse by small margin, so we ultimately decided this augmentation wasn’t optimal. Our hypothesis is that structures like the ones described above help the model construct plausible crystals.
> > >
> > > **Definition of stable and metastable**
> > >
> > > We defined a bulk material as stable if $\hat{E}_{\text{hull}} < 0.0$ and metastable if $< 0.1$. Stability corresponds to being below the energy hull (a very restrictive condition), while metastability just corresponds with being near the energy hull (a less restrictive condition).
> > >
> > > **Closing comments**
> > >
> > > We hope that the details provided above help address the questions in your review. If you have remaining questions, we would be happy to discuss any facet of the paper further. We have put a significant effort into this response, including several new details. In light of this response, we hope you will consider raising your score.
> > >
> > > **References**
> > >
> > > [1] Tian Xie, Xiang Fu, Octavian-Eugen Ganea, Regina Barzilay, and Tommi Jaakkola. Crystal diffusion variational autoencoder for periodic material generation. arXiv preprint arXiv:2110.06197, 2021.
> > >
> > > [2] Jiao, Rui, et al. "Crystal structure prediction by joint equivariant diffusion." arXiv preprint arXiv:2309.04475 (2023).
> > >
> > > [3] Daniel Flam-Shepherd and Alán Aspuru-Guzik. Language models can generate molecules, materials, and protein binding sites directly in three dimensions as xyz, cif, and pdb files. arXiv preprint arXiv:2305.05708, 2023.
> > >
> > > [4] Luis M Antunes, Keith T Butler, and Ricardo Grau-Crespo. Crystal structure generation with autoregressive large language modeling. arXiv preprint arXiv:2307.04340, 2023.
> > >
> > > [5] Jiang, Albert Q., et al. "Mistral 7B." arXiv preprint arXiv:2310.06825 (2023).
> > >
> > > [6] Sanchez, Guillaume, et al. "Stay on topic with classifier-free guidance." arXiv preprint arXiv:2306.17806 (2023)
> > >
> > > [7] https://huggingface.co/docs/transformers/internal/generation_utils.
> > >
> > > [8] Jain, Anubhav, et al. "Commentary: The Materials Project: A materials genome approach to accelerating materials innovation." APL materials 1.1 (2013).

---

> > > > ### Comment · Reviewer_1Fky · 2023-11-22
> > > >
> > > > I thank the authors for their response. But I'm still confused about the atomic order problem. As the example shows, if the atomic order changes, such as
> > > > ```
> > > > Lattice
> > > >     abc: 6.29 6.29 6.34
> > > >     angles: 60.3 60.3 60.0
> > > > Coordinates (fractional)
> > > >     Pb (0.0, 0.0, 0.0)
> > > >     Pd (0.5, 0.5, 0.0)
> > > >     Pd (0.5, 0.0, 0.0)
> > > >     Pd (0.0, 0.5, 0.0)
> > > >     Pb (0.5, 0.5, 0.5)
> > > >     Se (0.21, 0.21, 0.36)
> > > >     Se (0.79, 0.79, 0.64)
> > > > ```
> > > > The two systems are indeedly same. Will the framework recognize it?

---

> ### Author Response · Authors · 2023-11-22
>
> Thank you for your follow-up question. To clarify, the two examples are not identical to our LLMs. The example you provided and the example that we provided will be assigned slightly different likelihoods. We found that encouraging the model to learn an exact invariance hurt overall sample quality, most likely for the reasons we explore in the rebuttal. If we look at the likelihood that the model assigns to each of these examples, however, we see that it is very nearly invariant to these types of changes (permutations of the ordering) by default. To demonstrate this invariance, let's consider two types of changes
>
> **(a) permutations of the ordering**, for example
>
> ```
> Lattice
>     abc: 6.29 6.29 6.34
>     angles: 60.3 60.3 60.0
> Coordinates (fractional)
>     Pd (0.5, 0.5, 0.0)
>     Pd (0.5, 0.0, 0.0)
>     Pd (0.0, 0.5, 0.0)
>     Pb (0.5, 0.5, 0.5)
>     Pb (0.0, 0.0, 0.0)
>     Se (0.21, 0.21, 0.36)
>     Se (0.79, 0.79, 0.64)
> ```
> to
> ```
> Lattice
>     abc: 6.29 6.29 6.34
>     angles: 60.3 60.3 60.0
> Coordinates (fractional)
>     Pb (0.0, 0.0, 0.0)
>     Pd (0.5, 0.5, 0.0)
>     Pd (0.5, 0.0, 0.0)
>     Pd (0.0, 0.5, 0.0)
>     Pb (0.5, 0.5, 0.5)
>     Se (0.21, 0.21, 0.36)
>     Se (0.79, 0.79, 0.64)
> ```
>
> **(b) permutations of the ordering and swapping elements**, for example
>
> ```
> Lattice
>     abc: 6.29 6.29 6.34
>     angles: 60.3 60.3 60.0
> Coordinates (fractional)
>     Pd (0.5, 0.5, 0.0)
>     Pd (0.5, 0.0, 0.0)
>     Pd (0.0, 0.5, 0.0)
>     Pb (0.5, 0.5, 0.5)
>     Pb (0.0, 0.0, 0.0)
>     Se (0.21, 0.21, 0.36)
>     Se (0.79, 0.79, 0.64)
> ```
> to
> ```
> Lattice
>     abc: 6.29 6.29 6.34
>     angles: 60.3 60.3 60.0
> Coordinates (fractional)
>     Se (0.5, 0.5, 0.0)
>     Se (0.5, 0.0, 0.0)
>     Se (0.0, 0.5, 0.0)
>     Pb (0.21, 0.21, 0.36)
>     Pb (0.79, 0.79, 0.64)
>     Pd (0.5, 0.5, 0.5)
>     Pd (0.0, 0.0, 0.0)
> ```
>
> Transformations of type **(a)** should be unimportant to the model while changes of type **(b)** can generate completely implausible structures and should therefore decrease the likelihood assigned by the model. If we sample 10 transformations of each type for the original example and obtain the log likelihoods of each transformed structure as assigned by LLaMA-2 70B we obtain the following average changes in log likelihood (from the starting example to the transformed example).
>
> | Transformation Type | Avg. Delta Log Likelihood |
> | -------- | -------- |
> | (a) | -0.077 |
> | (b) | -3.647 |
>
> For permutations of the ordering, the log likelihood barely changes, while changes that swap element identities typically decrease the log likelihood significantly. Thus, while we do not explicitly enforce permutation invariance, we can see that our model learns that permutations are not very significant and that other types of transformations are.

---

### Official Review · Reviewer_iXqY · 2023-10-30

**Soundness:** 2 fair
**Presentation:** 3 good
**Contribution:** 2 fair
**Rating:** 5
**Confidence:** 3

**Summary:**

This paper propose a new application area for large language models (LLMs), i.e. LLaMA-2. The author leverages parameter-efficient fine-tuning to use LLMs. Based on domain expertise in material sciences, the author evaluated the proper tokenization methods for crystal structures and developed new metric to further include symmetric information into fine-tuning process.

**Strengths:**

1. the author got good performance in the shown benchmarks.
2. the method seems solid since it has been widely used in many other domains.

**Weaknesses:**

1. While one anticipates good performance from LLMs on standard evaluation metrics, especially with the likes of LLaMA-70B, the critical matter lies in the practical application in experiments.

2. Given that LLMs can sometimes produce hallucinations, it would be beneficial to comprehensively evaluate this behavior in large models, rather than merely touching upon it in the limitations section. Presenting failure examples could offer valuable insights.

3. Regarding Figure 2, could the authors elucidate why larger models result in poorer coverage?

**Questions:**

See above.

---

> ### Author Response · Authors · 2023-11-20
> **Author Rebuttal to Reviewer iXqY**
>
> Thank you for your review. We engage with each of your questions individually below:
>
> > “one anticipates good performance from LLMs on standard evaluation metrics, especially with the likes of LLaMA-70B”
>
> Although LLMs and parameter-efficient finetuning have become popular for many tasks in vision and natural language processing, they are not commonly applied to problems in materials science, a field that typically does not typically utilize foundation models [1]. Our application is very nonstandard for language models and many researchers would not expect language models to perform well on this task because they are not designed with modeling 3D space in mind. Our paper is surprising and exciting because it shows that text pretraining allows LLMs to model physical constraints very effectively, with a small number of trainable parameters on par with domain-specific models trained from scratch (4.5 million trainable parameters for CDVAE [2] vs. 3.5 million trainable parameters for LLaMA-2 7B with LoRA).
>
> >"the critical matter lies in the practical application in experiments"
>
> We strongly agree that practical performance is critical, and this belief was the motivation behind our extensive evaluations with DFT. These simulations verify that the model can generate plausible materials at a much higher rate than competing approaches and therefore requires less filtering before deployment in real physical experiments. The fact that LLMs consistently generate stable materials is very strong evidence that LLMs are learning fundamental properties of materials and not simply superficial correlations. The computational burden of our methods is also not prohibitive, as we detail in our general rebuttal note. Generation from our smallest LLaMA model requires only about 24 mins for 10,000 samples, while achieving better rates of metastability under M3GNet and comparable rates of stability under DFT. There are countless teams worldwide deploying real applications on top of LLaMA models of all sizes, and LLaMA-2 70B was downloaded 177,000 times from Hugging Face last month alone. Use of these models, therefore, doesn’t preclude practical applications and can in fact make it easier to leverage general advances in training and deploying LLMs (e.g. the GPT-4 fine-tuning API).
>
> > “Presenting failure examples could offer valuable insights”
>
> We first note that there is already a hallucination example displayed in Figure 4, which is discussed on page 8. In this example, the model hallucinates an element named “Ln”. There is no such element on the periodic table, but “Ln” is a common abbreviation for Lanthanide, a class of elements on the periodic table. Much like other hallucinations, this generation is plausible but ultimately incorrect.
>
> > “Given that LLMs can sometimes produce hallucinations, it would be beneficial to comprehensively evaluate this behavior in large models”
>
> While hallucinations are challenging in many LLMs applications because of the challenges of verifying correct answers, we would like to clarify that the types of hallucinations described here are easy to deal with. There are two forms of hallucination that occur in our case: (a) hallucination of non-physical structures or compositions (b) hallucination of non-existent elements. As we show in the paper, hallucination of non-physical compositions actually occurs at a lower rate in our models compared to the baselines and the rate of non-physical structures approaches zero. Because we use LLaMA-2 tokens without modification, it is possible for the model to sample imaginary elements, but we found that this behavior became exceedingly rare in large models, and could be avoided altogether by only sampling tokens corresponding to known elements. In both cases, samples can be checked for validity in less than one second, making hallucinations in materials generation fundamentally different than in many other LLM applications.
>
> It is also worth noting that while the term “hallucination” was coined for LLMs, unlikely or inconsistent samples can be an issue for all generative models [3]. For example, CDVAE commonly hallucinates combinations of elements that are physically implausible.

---

> ### Author Response · Authors · 2023-11-20
> **Author Rebuttal to Reviewer iXqY (part 2)**
>
> We believe our paper has many compelling and timely contributions. First and foremost, we observe near-perfect rates of validity and state-of-the-art rates of stability under DFT simulations. As we outline in the general rebuttal comment, the computation cost of obtaining state-of-the-art samples is on par with prior work, making the approach practically relevant. The use of pre-trained LLMs also unlocks an entirely new set of possibilities for generating crystals conditioned on text descriptions. There were several key methodological details that were carefully engineered, including the crystal formatting and data augmentation method. With an optimal formatting strategy, our method is easy to extend to any LLM, including systems like GPT-4. We hope that our work will demonstrate that LLMs are in fact incredibly general-purpose generative models and that their impact can extend to many tasks within materials science and chemistry.
>
> We would welcome any further questions, and we hope that, in light of our detailed response and clarifications, you will consider raising your score to support acceptance of the paper.
>
> **References**
>
> [1] Zhang, Xuan, et al. "Artificial intelligence for science in quantum, atomistic, and continuum systems." arXiv preprint arXiv:2307.08423 (2023).
>
> [2] Tian Xie, Xiang Fu, Octavian-Eugen Ganea, Regina Barzilay, and Tommi Jaakkola. Crystal diffusion variational autoencoder for periodic material generation. arXiv preprint arXiv:2110.06197, 2021.
>
> [3] Ji, Ziwei, et al. "Survey of hallucination in natural language generation." ACM Computing Surveys 55.12 (2023): 1-38.

---

### Official Review · Reviewer_9VCZ · 2023-11-08

**Soundness:** 3 good
**Presentation:** 3 good
**Contribution:** 2 fair
**Rating:** 6
**Confidence:** 3

**Summary:**

This paper proposed to use LLMs to model the materials. The main technical contribution comes from 1) a modified tokenization mechanism that is suitable for material data; 2) a data augmentation method that leverages the property of the materials; 3) the prompt designs. Experiments on benchmark datasets show that when tuned on LLAMA2, it is able to improve the performance compared to other baselines.

**Strengths:**

- Application on the material science domain is interesting
- The adaptation of LLMs for the material science is reasonable
- Empirical results are promising

**Weaknesses:**

- The technical contribution is relatively limited, where the tokenization, the prompt design and the objectives for pertaining are all well studied in the past.
- Since the nature of the material data is quite different compared to the natural language data, I’m wondering whether the pretrained LLAMA2 is offering any additional value. It would be helpful if one can show the performance with and without loading the pretrained checkpoint from LLAMA2 tasks.

**Questions:**

I’d like to see the author’s response on the questions I listed in the weakness section.

---

> ### Author Response · Authors · 2023-11-20
> **Author Rebuttal to Reviewer 9VCZ**
>
> Thank you for your review. We would like to clarify that text pretraining is essential to our method for two reasons.
> 1. It would be impractically expensive or computationally infeasible to train models with up to 70B parameters from scratch on our data. Using a pretrained model with LoRA [1] offers the benefits of model scale while maintaining tractability and limiting overfitting, as the actual number of trainable parameters can be relatively small.
>
> 2. Pretraining on text data yields a model that can be conditioned on text for free, and text conditioning opens up a huge new realm of exciting possibilities, like conditioning samples on desired properties. It would be challenging to achieve a similar result from scratch without significantly expanding the size of the dataset (to improve general text understanding) and without essentially training a general-purpose language model in the process.
>
> To better understand the first point, let’s quickly review the exact details of the finetuning procedure. We are using low-rank adapters (LoRA), as opposed to end-to-end finetuning, and this means we are adding a small number of additional parameters to an existing, frozen model. The easiest way to see the difference between this approach and training a model from scratch (as in [2]) is to compare the training loss over the first few epochs of training.
>
> | Model | Epoch 1 | Epoch 2 | Epoch 3 | Epoch 4 | Epoch 5 |
> | -------- | -------- | -------- | -------- | -------- | -------- |
> | GPT-2 (from scratch) | 0.946 | 0.878 | 0.807 | 0.757 | 0.740 |
> | LLaMA-13B (LoRA) | 0.457 | 0.432 | 0.424 | 0.401 | 0.385 |
> | LLaMA-70B (LoRA) | 0.402 | 0.344 | 0.325 | 0.305 | 0.296 |
>
> If we attempt to run LoRA finetuning with randomly initialized parameters for the LLaMA-2 7B model we observe an immediate and significant difference in the training losses:
>
> | Model | 1 Iter | 0.33 Epochs | 0.66 Epochs | 1 Epoch  |
> | -------- | -------- | -------- | -------- | -------- |
> | Random | 13.46 | 1.53 | 0.81 | 0.78 |
> | Pre-trained | 1.57 | 0.47 | 0.41 | 0.39 |
>
> While LoRA finetuning is tractable because 99.95% of the model is frozen, finetuning a LLaMA-2 model end-to-end in half-precision would require at least 4 times as many GPUs, making it infeasible for all but a handful of researchers. When using LoRA, even though the base models are large the number of trainable parameters is very small. In fact, the LLamA-2 7B model has less trainable parameters than one of the baseline methods we compared (CDVAE) [3]. The number of trainable parameters for each of our models and the baseline models is shown below:
>
> *CDVAE [3]*: 4.5 million trainable parameters (100% of total)
>
> *LM-CH/AC [2]*: 1-100 million trainable parameters (100% of total)
>
> *LLaMA-2 7B*: 3.5 million trainable parameters (0.05% of total)
>
> *LLaMA-2 13B*: 6.5 million trainable parameters (0.05% of total)
>
> *LLaMA-2 70B*: 35 million trainable parameters (0.05% of total)

---

> ### Author Response · Authors · 2023-11-20
> **Author Rebuttal to Reviewer 9VCZ (part 2)**
>
> Having clarified the significance of using pretraining, we now expand upon the unique facets of our method and their significance. While prompt design and tokenization are established research areas, they have not been extensively studied within generative models of materials, or generative models of molecules more broadly (e.g. protein structures). We cite two papers that use language models to sample novel materials, both of which were published within the last six months [2,4]. Neither of these papers explores prompt design, and only one ablates tokenization methods [2]. Both papers use models trained from scratch on domain-specific data.
>
> **Tokenization**
>
> Notably, our approach to tokenization is distinctly different from prior work on modeling atomic structures with language models. Instead of using a special vocabulary and training models from scratch, we use LLaMA-2’s existing tokenizer. This choice allows us to easily process both encoded crystals and text data. In early experiments, we tried out many other approaches, including fine-tuning LLaMA-2 models with additional tokens specific to crystal data. These methods were more challenging to train and didn’t lead to any improvements over using a shared tokenizer. We include a set of example training losses below:
>
> | | Epoch 1 | Epoch 2 | Epoch 3 | Epoch 4 | Epoch 5 |
> | -------- | -------- | -------- | -------- | -------- | -------- |
> | Special Crystal Tokens | 0.783 | 0.693 | 0.623 | 0.611 | 0.588 |
> | Shared Tokenization | 0.457 | 0.432 | 0.424 | 0.401 | 0.385 |
>
> **Prompt design**
>
> There are many important decisions involved both in text formatting (e.g the choice of fractional or absolute coordinates) and augmentation of the input data (e.g. translation or permutation augmentations on coordinates). We refined our choices over weeks of training and evaluation. As a simple example, we provide average validity numbers (using low temperature sampling) from earlier experiments on LLaMA-2 7B models trained different formatting styles
>
> | Setting | Structural Validity | Compositional Validity |
> | -------- | -------- | -------- |
> | Fractional coords | 0.914 | 0.832 |
> | Absolute coords  | 0.908  | 0.805 |
> | No permutations | 0.925 | 0.829 |
> | With permutations | 0.892  | 0.817 |
>
> Even the phrasing of the instruction prompt can have a significant impact on the ultimate rates of validity, for example by providing proper context for element symbols:
>
> | Setting | Structural Validity | Compositional Validity |
> | -------- | -------- | -------- |
> | Prompt 1 | 0.877 | 0.795 |
> | Prompt 2 | 0.929 | 0.839 |
> | Prompt 3 | 0.957 | 0.875 |
>
> Prompt 1: “Bulk material description:\n”
>
> Prompt 2: “Below is a description of a bulk material, starting with a list of atom types for the elemental composition followed by the lengths and angles of the lattice vectors and finally the atom type and coordinates for each atom within the lattice:\n\n"
>
> Prompt 3: "Please provide the description of a stable bulk material, starting with a list of atom types for the elemental composition followed by the lengths and angles of the lattice vectors and finally the atom type and coordinates for each atom within the lattice.\n\nAn example of a bulk material description is:\n\n{example}\n\nInclude the new material description below:\n\n"
>
> The length of Prompt 3 was prohibitive, so we ultimately abandoned it, despite promising results.
>
> **In closing**
>
> Ultimately, we think that simplicity is a virtue. Using our method, it is easy to adapt popular LLMs methods directly to improving generative models of materials. Because of the generality of our approach, it would also be straightforward to extend to other atomic structures, such as drug-like compounds, protein structures, or nucleic acids.
>
> We hope that our rebuttal addresses the concerns outlined in your review. If you have remaining questions, we would be happy to discuss any details further. We would appreciate it if you would consider raising your score in light of our response.
>
> **References**
>
> [1] J. Edward Hu, Yelong Shen, Phillip Wallis, Zeyuan Allen-Zhu, Yuanzhi Li, Shean Wang, and Weizhu Chen. Lora: Low-rank adaptation of large language models. ArXiv, abs/2106.09685, 2021.
>
> [2] Daniel Flam-Shepherd and Alán Aspuru-Guzik. Language models can generate molecules, materials, and protein binding sites directly in three dimensions as xyz, cif, and pdb files. arXiv preprint arXiv:2305.05708, 2023
>
> [3] Tian Xie, Xiang Fu, Octavian-Eugen Ganea, Regina Barzilay, and Tommi Jaakkola. Crystal diffusion variational autoencoder for periodic material generation. arXiv preprint arXiv:2110.06197, 2021.
>
> [4] Luis M Antunes, Keith T Butler, and Ricardo Grau-Crespo. Crystal structure generation with autoregressive large language modeling. arXiv preprint arXiv:2307.04340, 2023.

---

### Author Response · Authors · 2023-11-20
**Author Rebuttal**

We begin with a general comment, and then reply to reviewers individually as separate posts.

Although the scope of LLM research is rapidly expanding, it is not at all obvious that LLMs can be successful at the task of modeling atomic coordinates, or that their performance can exceed a purpose-build model operating directly on 3D coordinates with carefully-engineered symmetries constraints. The observation that text pretraining gives rise to biases that are relevant for modeling 3D structures is both fascinating and broadly impactful. Text pretraining allows us to finetune a model with roughly the same number of trainable parameters as a domain-specific model, and, perhaps more importantly, makes it unnecessary to compile a massive dataset of task-specific data in order to learn generalizable features of 3D structures.

Our use of a general-purpose tokenizer also makes it straightforward to expand the scope of our method beyond crystals to other atomic structures, such as drug-like compounds or protein structures and opens up an entirely new realm of possibilities in text-conditional generation and design, because our model can operate on mixtures of text and atomic structures.

Beyond demonstrating the novel capabilities of pretrained LLMs, we also demonstrate that our method generates a higher rate of stable structures than baselines. To do so, we move beyond basic validity statistics and perform full-blown DFT simulations. These simulations give us a high confidence that LLMs are learning generalizable features of bulk materials and that their samples are likely to be useful in actual materials discovery applications.

It’s most informative to compare our method with prior work along three primary axes:
- Capabilities: Our method can generate unconditional structures, infill existing structures or generate structures conditioned on text descriptions. Prior work [2,3] can only be used in the first of these three applications.
- Generality: Because our method uses a pre-trained model and LLaMA-2’s default tokenization, it can be easily extended to any string-formatted set of atomic coordinates. For example, a model could be trained jointly on inorganic crystals and organic molecules. This level of generality would be impossible for models with domain-specific architectures [2] and would be much more challenging when training a model from scratch [3].
- Performance: While unlocking new possibilities, our method also simply performs better than prior work when comparing rates of validity or stability.

| Model | Struct. Validity | Comp. Validity | Metastability (M3GNet) | Stability (DFT) |
| -------- | -------- | -------- | -------- | -------- |
| CDVAE | 100% | 86.7% | 28.8% | 5.4% |
| LM-CH | 84.8% | 83.5% | n/a | n/a |
| LM-AC | 95.8% | 88.9% | n/a | n/a |
| LLaMA-2 70B ($\tau$ = 1.0) | 96.5% | 86.3% | 35.4% | 10.0% |
| LLaMA-2 70B ($\tau$ = 0.7) | 99.6% | 95.4% | 49.8% | 10.6% |

---

> ### Author Response · Authors · 2023-11-20
> **Author Rebuttal (continued)**
>
> Lastly, we address the computational overhead of using our method. Although LLMs might seem like computational overkill at face value, if we perform a detailed analysis, we can see that deploying a LLaMA model for large-scale sampling of materials actually has comparable computational overhead to competing approaches. We can use LLaMA-2 13B and 70B as the models for our method and AWS as the deployment environment. In a recent benchmark on a cloud instance with 8 A100 GPUs (ml.p4d.12xlarge) [1], LLaMA-2 13B achieved 0.416 hr/1M tokens and LLaMA-2 70B achieved 0.864 hr/1M tokens. One crystal is around 100 tokens on average, so the throughput for 10,000 crystals is the same as for 1M tokens. For comparison, we use CDVAE and its recorded runtimes for generating 10,000 crystals on a single RTX2080 Ti GPU [2]. To obtain the final numbers, we adjust for the number of GPUs (8) and a 2x improvement from RTX2080 Ti to A100 GPUs [4].
>
> | Model | Hours / 10,000 crystals | Hours / 10,000 metastable (M3GNet) crystals |
> | -------- | -------- | -------- |
> | CDVAE | 0.363 | 1.260 |
> | LLaMA-13B | 0.416 | 1.094 |
> | LLaMA-70B | 0.864 | 1.728 |
>
> We see that LLaMA-2 13B actually has a comparable computational overhead to prior work, and LLaMA-2 70B is only slightly higher. When considering the rate of stable materials generated by each method, we see that LLaMA-2 13B actually has a higher throughput than CDVAE and with noticeably higher diversity. We think LLMs are particularly exciting because their ability to understand physical laws improved consistently with the performance of the base model, capping out at overall validity (structure and composition) rates near 90%. As models continue to become more efficient, LLMs are well-positioned to be both the best at generating valid structures and a computationally efficient option.
>
> We have additionally provided several experimental results inspired by your reviewer comments. We hope that the timeliness and significance of our work, these results, and our responses, can be considered in the final assessment.
>
> **References**
>
> [1] LLaMA 2 on Amazon Sagemaker, a Benchmark https://huggingface.co/blog/llama-sagemaker-benchmark
>
> [2] Tian Xie, Xiang Fu, Octavian-Eugen Ganea, Regina Barzilay, and Tommi Jaakkola. Crystal diffusion variational autoencoder for periodic material generation. arXiv preprint arXiv:2110.06197, 2021.
>
> [3] Daniel Flam-Shepherd and Alán Aspuru-Guzik. Language models can generate molecules, materials, and protein binding sites directly in three dimensions as xyz, cif, and pdb files. arXiv preprint arXiv:2305.05708, 2023
>
> [4] NVIDIA A100 GPU Benchmarks for Deep Learning https://lambdalabs.com/blog/nvidia-a100-gpu-deep-learning-benchmarks-and-architectural-overview

---

### Meta-Review · Area_Chair_zb5D · 2023-12-14

**Metareview:**

This paper explores the use of fine-tuning LLMs for generating atomic coordinates of stable crystal structures. The end results are quite impressive relative to existing baselines (e.g. diffusion models over coordinates), and the topic is quite timely. Reviewers had some concerns regarding the overall novelty and scope of the technical contribution (which focused primarily on tokenization, pretraining tasks, and prompt design), but the careful presentation and evaluation of the work, as well as its relevance and strong overall performance, lean on balance towards acceptance.

**Justification For Why Not Higher Score:**

Technical contribution is somewhat straightfoward

**Justification For Why Not Lower Score:**

Performance is very good, presentation is clear, establishes a strong LLM baseline to compare against other special-purpose methods

---

### Decision · Program_Chairs · 2024-01-16

Accept (poster)